# Turbo Learning for CaptionBot and DrawingBot

**Qiuyuan Huang**
Microsoft Research
Redmond, WA, USA
qihua@microsoft.com

**Pengchuan Zhang**
Microsoft Research
Redmond, WA, USA
penzhan@microsoft.com

**Dapeng Wu**
University of Florida
Gainesville, FL, USA
dpwu@ieee.org

**Lei Zhang**
Microsoft Research
Redmond, WA, USA
leizhang@microsoft.com

## Abstract

We study in this paper the problems of both image captioning and text-to-image generation, and present a novel turbo learning approach to jointly training an image-to-text generator (a.k.a. CaptionBot) and a text-to-image generator (a.k.a. DrawingBot). The key idea behind the joint training is that image-to-text generation and text-to-image generation as dual problems can form a closed loop to provide informative feedback to each other. Based on such feedback, we introduce a new loss metric by comparing the original input with the output produced by the closed loop. In addition to the old loss metrics used in CaptionBot and DrawingBot, this extra loss metric makes the jointly trained CaptionBot and DrawingBot better than the separately trained CaptionBot and DrawingBot. Furthermore, the turbo-learning approach enables semi-supervised learning since the closed loop can provide pseudo-labels for unlabeled samples. Experimental results on the COCO dataset demonstrate that the proposed turbo learning can significantly improve the performance of both CaptionBot and DrawingBot by a large margin.

## 1   Introduction

Due to the breakthrough of deep learning, recent years have witnessed great progresses in both computer vision and natural language processing. As a result, two fundamental problems – image captioning and text-to-image generation – that requires cross modality understanding have also been intensely studied in the past few years. Image captioning (a.k.a. CaptionBot) is to generate a meaningful caption for any given input image, whereas text-to-image (a.k.a. DrawingBot) is to generate a realistic image for any input sentence. Regardless of their different implementations, it is interesting to see that the two problems can be regarded as dual problems as both can take each other's output as input. However, despite their duality in terms of their input and output forms, the two problems were largely studied separately in the past, leaving a significant room for improvement.

To leverage their duality, for the first time, this paper proposes a turbo-learning approach, which can jointly train a CaptionBot and a DrawingBot together in a way similar to an engine turbo-charger, which feeds the output back to the input to reuse the exhaust gas for increased back pressure and better engine efficiency. The key idea of the proposed turbo-learning approach is that an image-to-text generator and a text-to-image generator can form a closed loop and generate informative feedback signals to each other. In this way, we can obtain a new loss metric (for updating the neural network weights during training) by comparing the original input data with the output data produced by the whole closed loop, in addition to the old loss metric which measures the difference between the output and the ground-truth of the CaptionBot or the DrawingBot. This extra loss metric effectively

leads to improved performance of both of the jointly trained CaptionBot and DrawingBot compared with their separately trained counterparts.

To jointly train a CaptionBot and a DrawingBot, we utilize the state-of-the-art long short-term memory (LSTM) image captioner [1] and text-to-image generation algorithm [2] as building blocks and use stochastic gradient descent to iteratively learn the network parameters of both blocks. More specifically, as illustrated in Fig. 1, training in each iteration consists of two steps. In Step 1, the DrawingBot serves as the primal module while the CaptionBot serves as the dual module. In Step 2, the CaptionBot serves as the primal module while the DrawingBot serves as the dual module. In each step, the primal module takes an input and produces/forwards its output to its dual module, and the dual module further feeds back its output to the input of the primal module. The weights of the DrawingBot and the CaptionBot are updated simultaneously in each step by using the loss function of two pairs – (gold image vs. generated image) and (gold sentence vs. generated sentence) to calculate the gradient.

Closing the loop between CaptionBot and DrawingBot also brings another merit: it enables semi-supervised learning since the closed loop can provide pseudo-labels for unlabeled samples. For image captioning, such a semi-supervised learning capability is particularly useful since human annotation of images is very costly. Note that between the two bots in each step we also have a constraint to ensure the sentence (or image) generated from the primal module is natural (or realistic). The constraint could be a Generative Adversarial Network (GAN) loss if we only have unsupervised training data, i.e., images and sentences without correspondence. In this work, we use the semi-supervised setting for simplicity and use the ground truth sentence (or image) to supervise/constrain the sentence (or image) generated from the primal module, while letting the unlabeled data to pass through the two modules to improve the feature learning. This loss term effectively prevents the model from learning trivial identity mappings for both the CaptionBot and DrawingBot.

We conducted experimental evaluation on the COCO dataset. Experimental results show that the turbo learning approach significantly improves the performance of CaptionBot and DrawingBot. For example, under supervised learning, the CaptionBot is improved by 13% in BLEU-4 metric, and the DrawingBot is improved by 4.3% in Inception score; under semi-supervised learning, the CaptionBot is improved by 77% in BLEU-4 metric.

The rest of the paper is organized as follows. Section 2 discusses the related work. In Section 3, we introduce our proposed turbo learning approach and present its theoretical underpinning. Section 4 describes turbo learning for supervised training and presents experimental results. In Section 5, we present our semi-supervised learning for CaptionBot and experimental results. Finally, Section 6 concludes the paper.

## 2   Related work

Most existing image captioning systems exploit end-to-end deep learning with a convolutional neural network (CNN) image-analysis front end producing a distributed representation that is then used to drive a natural-language generation process, typically using a recurrent neural network (RNN) or LSTM [3, 4, 5, 6, 7, 8, 9, 10, 11, 12, 13, 14].

Generating photo-realistic images from text descriptions has been an active research area in recent years. There are different approaches working toward this grand goal, such as variational inference [15, 16], approximate Langevin process [17], conditional PixelCNN via maximal likelihood estimation [18, 17], and conditional generative adversarial networks [19, 20, 21, 22]. Compared with other approaches, generative adversarial networks (GANs) [23] have shown great performance for generating sharper samples [24, 25, 26, 27, 28]. AttnGAN [2] proposes an attentional multi-stage generator that can synthesize fine-grained details at different image regions by paying attentions to the relevant words in the text description, which achieves the state-of-the-art performance of the text-to-image synthesis task on MS-COCO dataset.

Different from previous works which study the problems of image captioning and text-to-image generation separately, this work aims at developing a turbo learning approach to jointly training CaptionBot and DrawingBot. Our proposed turbo learning is similar to dual learning [29], which is applied to automatic machine translation. Under dual learning, any machine translation task has a dual task, e.g., English-to-French translation (primal) versus French-to-English translation (dual); the

primal and dual tasks can form a closed loop, and generate feedback signals to train the translation models, even if without the involvement of a human labeler. In the dual-learning mechanism, one agent represents the model for the primal task and the other agent represents the model for the dual task, then the two agents teach each other through a reinforcement learning process. Dual-learning is solely developed for NLP. In contrast, our turbo learning is applied to both NLP and computer vision, which is more challenging.

# 3 Turbo learning structure

In this section, we present the turbo learning approach for training CaptionBot and DrawingBot jointly.

## 3.1 Turbo learning for CaptionBot and DrawingBot

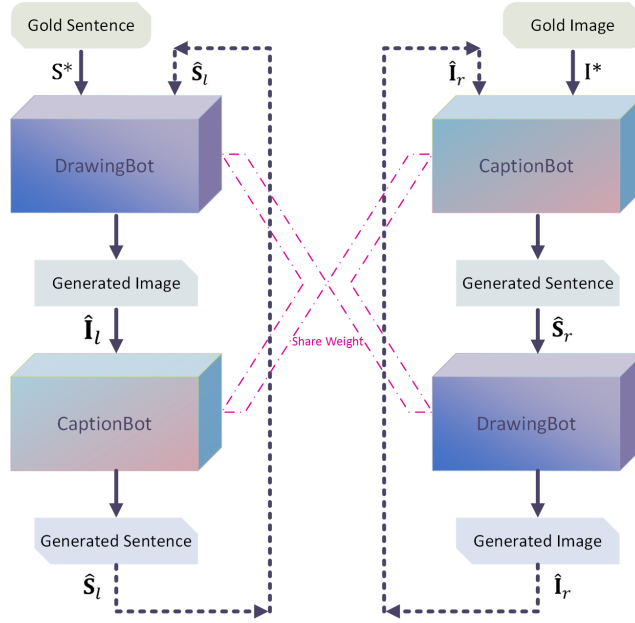

Figure 1: The proposed turbo butterfly structure for training CaptionBot and DrawingBot jointly

Fig. 1 shows the turbo architecture for the proposed joint training of CaptionBot and DrawingBot. As shown in the left hand side of Fig. 1, for a given gold sentence $\mathbf{S}^*$, the DrawingBot generates an image $\hat{\mathbf{I}}_l$; the generated image is supplied to the CaptionBot, which produces a captioning sentence $\hat{\mathbf{S}}_l$. Then we use the loss function of pairs $(\hat{\mathbf{S}}_l, \mathbf{S}^*)$ and $(\hat{\mathbf{I}}_l, \mathbf{I}^*)$ to calculate the gradients and update $\theta_{draw}$ and $\theta_{cap}$, the parameters of the DrawingBot and CaptionBot, simultaneously. As shown in the right hand side of Fig. 1, for a given gold image $\mathbf{I}^*$, the CaptionBot generates a sentence $\hat{\mathbf{S}}_r$; the generated sentence is supplied to the DrawingBot, which produces an image $\hat{\mathbf{I}}_r$. Then we use the loss function of pairs $(\hat{\mathbf{S}}_r, \mathbf{S}^*)$ and $(\hat{\mathbf{I}}_r, \mathbf{I}^*)$ to calculate the gradients and update $\theta_{draw}$ and $\theta_{cap}$ simultaneously.

## 3.2 Optimization theoretical underpinning of turbo learning

Considering the left hand side of Fig. 1, for a given gold sentence $\mathbf{S}^*$, the DrawingBot generates an image $\hat{\mathbf{I}}_l$; mathematically, we can represent this process by

$$\hat{\mathbf{I}}_l = \mathcal{D}(\mathbf{S}^*|\theta_{draw}) \tag{1}$$

where $\mathcal{D}(\cdot)$ denotes the DrawingBot as a nonlinear operator parameterized by $\theta_{draw}$. The generated image $\hat{\mathbf{I}}_l$ is supplied to the CaptionBot, which produces a captioning sentence $\hat{\mathbf{S}}_l$; mathematically, we

can represent this process by

$$\hat{\mathbf{S}}_l = \mathcal{C}(\hat{\mathbf{I}}_l | \theta_{cap}) \tag{2}$$

where $\mathcal{C}(\cdot)$ denotes the CaptionBot as a nonlinear operator parameterized by $\theta_{cap}$; conceptually, operator $\mathcal{C}(\cdot)$ can be considered as an approximate inverse of operator $\mathcal{D}(\cdot)$. Similarly in the right branch, we have $\hat{\mathbf{S}}_r = \mathcal{C}(\mathbf{I}^* | \theta_{cap})$ and $\hat{\mathbf{I}}_r = \mathcal{D}(\hat{\mathbf{S}}_r | \theta_{draw})$.

Ideally, we intend to minimize both $\mathbb{E}_{(\mathbf{S}^*, \mathbf{I}^*)}[L_l(\theta_{draw}, \theta_{cap})]$ and $\mathbb{E}_{(\mathbf{S}^*, \mathbf{I}^*)}[L_r(\theta_{draw}, \theta_{cap})]$, in which $L_l(\theta_{draw}, \theta_{cap})$ and $L_r(\theta_{draw}, \theta_{cap})$ are the loss function of the left hand side and the right hand side of Fig. 1, respectively. We solve this multi-objective optimization problem by converting it to a single objective optimization problem by summing up these two loss functions, i.e.,

$$\min_{\theta_{draw}, \theta_{cap}} \mathbb{E}_{(\mathbf{S}^*, \mathbf{I}^*)}[L_l(\theta_{draw}, \theta_{cap})] + \mathbb{E}_{(\mathbf{S}^*, \mathbf{I}^*)}[L_r(\theta_{draw}, \theta_{cap})]. \tag{3}$$

Our turbo training is intended to solve the optimization problem in (3) by the stochastic gradient descent algorithm, where we randomly select one of the two terms and approximate the expectation by an empirical average over a mini-batch. In practice, we select these two terms alternatively, and use the same mini-batch to compute the gradient for both $L_l$ and $L_r$. This reuse of samples cuts the I/O cost by half during the training.

### 3.3 Insight of turbo learning

Later in Section 4.4, the experimental results show that our proposed turbo joint training of CaptionBot and DrawingBot achieves significant gains over separate training of CaptionBot and DrawingBot. The reason is as follows. If we train CaptionBot and DrawingBot jointly under the turbo approach, the turbo CaptionBot has both sentence information (generated sentence $\hat{\mathbf{S}}$ vs. gold sentence $\mathbf{S}^*$) and image information (generated image $\hat{\mathbf{I}}$ vs. gold image $\mathbf{I}^*$) to guide the update of $\theta_{cap}$; in other words, the turbo CaptionBot is trained to generate a sentence, which not only is close to the ground truth sentence, but also captures as many classes/objects in the original input image as possible. In contrast, the baseline CaptionBot only has sentence information (generated sentence $\hat{\mathbf{S}}$ vs. gold sentence $\mathbf{S}^*$) for updating $\theta_{cap}$. The same reasoning is also true for the turbo DrawingBot.

## 4 Turbo learning for supervised training

This section is organized as follows. Subsection 4.1 and Subsection 4.2 describe CaptionBot and DrawingBot, respectively; Subsection 4.3 describes turbo joint training of CaptionBot and Drawing-Bot. Subsection 4.4 shows the experimental results.

### 4.1 LSTM model for CaptionBot

To illustrate the benefit of the turbo structure, in this section, we choose a simple baseline LSTM model as shown in Fig. 2 for the CaptionBot .

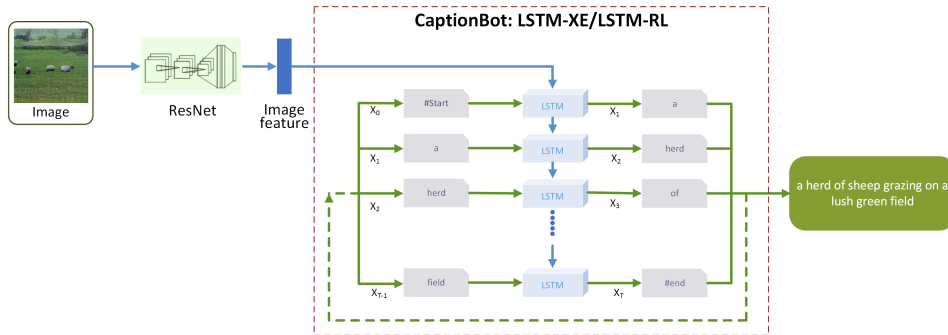

Figure 2: Baseline LSTM model for CaptionBot

Consider an image $\mathbf{I}$ with caption $\mathbf{S}$. Assume that caption $\mathbf{S}$ consists of $T$ words. We define $\mathbf{S} = [\mathbf{x}_1, \cdots, \mathbf{x}_T]$, where $\mathbf{x}_t$ is a one-hot encoding vector of dimension $V$, where $V$ is the size of the

vocabulary. The length $T$ may be different for different captions. The $t$-th word in a caption, $\mathbf{x}_t$, is embedded into an $n_x$-dimensional real-valued vector $\mathbf{w}_t = \mathbf{W}_e \mathbf{x}_t$, where $\mathbf{W}_e \in \mathbb{R}^{n_x \times V}$ is a word embedding matrix.

The following is the basic LSTM model for captioning a single image which is first processed by a Convolutional Neural Network (CNN) such as ResNet and then conveyed to the LSTM [30, 31, 1].

$$\mathbf{h}_t, \mathbf{c}_t = LSTM(\mathbf{x}_{t-1}, \mathbf{h}_{t-1}, \mathbf{c}_{t-1}) \tag{4}$$

where $\mathbf{h}_t, \mathbf{c}_t$ are the hidden state and the cell state of the LSTM at time $t$, respectively.

The LSTM model in Eq. (4) for image captioning can be trained via minimizing the cross-entropy (XE) loss function, which is called Baseline LSTM-XE in this paper.

To directly optimize NLP metrics such as CIDEr [32] and address the exposure bias issue, reinforcement learning (RL) can be used to train the LSTM model in Eq. (4) [12], which is called Baseline LSTM-RL in this paper. Under the RL terminology, the LSTM is regarded as an "agent" that interacts with an external "environment" (e.g., words and image features). The parameters of the LSTM, $\theta$, define a policy $p_\theta$, that produces an "action" (e.g., prediction of the current word). After each action, the agent updates its internal "state" (e.g., cell and hidden states of the LSTM). Upon generating the end-of-sentence (EOS) symbol, the agent receives a "reward" denoted as $r(S)$ (e.g., the CIDEr metric of the generated sentence w.r.t. the ground-truth sentence), where $S$ is the generated sentence.

Minimizing the reinforcement loss (or maximizing rewards) does not ensure the readability and fluency of the generated caption [33]. Using a mixed loss function, which is a weighted combination of the cross-entropy (XE) loss $L_{XE}$ and the reinforcement learning (RL) loss $L_{RL}$, helps improve readability and fluency since the cross-entropy loss is based on a conditioned language model, with a goal of producing fluent captions. That is, the mixed loss function is given by

$$L_{mix}(\theta) = \gamma L_{RL}(\theta) + (1 - \gamma) L_{XE}(\theta) \tag{5}$$

where $\gamma \in [0, 1]$.

## 4.2 AttnGAN for DrawingBot

AttnGAN [2] introduces an attentional multi-stage generative network, which can synthesize fine-grained details at different sub-regions of the image by paying attentions to the relevant words in the natural language description. Trained by multi-level discriminators and a deep attentional multimodal similarity model (DAMSM) that computes a fine-grained image-text matching loss, the AttnGAN achieves the state-of-the-art performance on the text-to-image generation tasks. In this work, we use AttnGAN, more precisely, its attentional multi-stage generative network, as the drawing bot.

As shown in Figure 3, the proposed attentional generative network has $m$ stages $F_i$ ($i = 0, 1, \ldots, m$), each of which outputs an hidden state $h_i$. On the last stage, we obtain the generated high-resolution image after passing the last hidden state $h_{m-1}$ through a convolutional layer with $3 \times 3$ kernel size. Specifically, the drawing bot, denoted as $\hat{\mathbf{I}} = \mathcal{C}(\mathbf{S}, z)$, is decomposed as

$$
\begin{aligned}
h_0 &= F_0(z, F^{ca}(\overline{e})); \\
h_i &= F_i(h_{i-1}, F_i^{attn}(e, h_{i-1})) \text{ for } i = 1, 2, ..., m - 1; \\
\hat{\mathbf{I}} &= G_{m-1}(h_{m-1}).
\end{aligned}
\tag{6}
$$

Here, $z$ is a noise vector usually sampled from a standard normal distribution. $\overline{e}$ is the global sentence vector of input sentence $\mathbf{S}$, and $e$ is the matrix of word vectors. The embedding was pretrained by the DAMSM model proposed by [2]. $F^{ca}$ represents the Conditioning Augmentation [21] that converts the sentence vector $\overline{e}$ to the conditioning vector. $F_i^{attn}$ is the proposed attention model at the $i^{th}$ stage in [2]. $F^{ca}$, $F_i^{attn}$, $F_i$, and $G_{m-1}$ are all modeled as neural networks. The AttnGAN is trained by minimizing the sum of 1) the GAN matching loss that jointly approximates conditional and unconditional distributions of multi-scale images, and 2) a word-level image-text matching loss.

## 4.3 Turbo training procedure for supervised learning

Now, we describe the proposed turbo training procedure. As mentioned before, we select $L_l(\theta_{draw}, \theta_{cap})$ and $L_r(\theta_{draw}, \theta_{cap})$ alternatively, i.e., minimizing $L_l(\theta_{draw}, \theta_{cap})$ and $L_r(\theta_{draw}, \theta_{cap})$ alternatively. Hence, each iteration of the turbo training procedure consists of the following three steps:

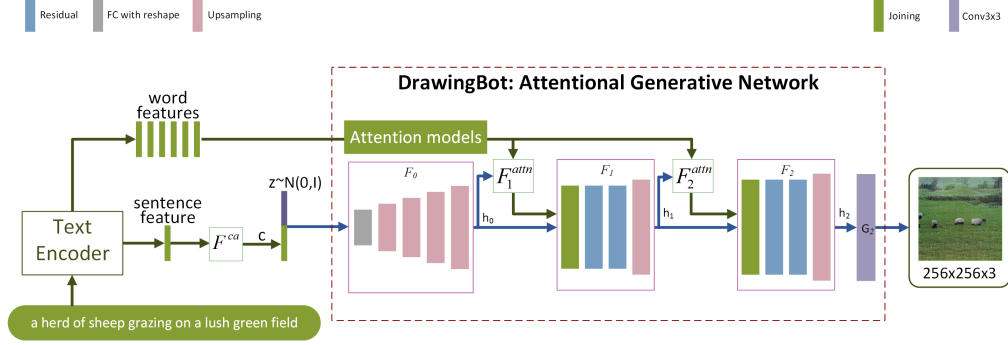

Figure 3: Drawingbot: the attentional multi-stage generative model from AttnGAN [2]

- **Step 1: minimizing** $L_l(\theta_{draw}, \theta_{cap})$. As shown in the left hand side of Fig. 1, for a given gold sentence $\mathbf{S}^*$, the DrawingBot generates an image $\hat{\mathbf{I}}_l$; the generated image is supplied to the CaptionBot, which produces a captioning sentence $\hat{\mathbf{S}}_l$. Then we use the following loss function to calculate the gradients and update $\theta_{draw}$ and $\theta_{cap}$ simultaneously:

$$L_l(\theta_{draw}, \theta_{cap}) = \beta_2(\alpha(\hat{\mathbf{I}}_l, \mathbf{I}^*) - r(\hat{\mathbf{S}}_l)) + (1 - \beta_2)(\beta_1\alpha(\hat{\mathbf{I}}_l, \mathbf{I}^*)) + (1 - \beta_1)L_{XE}(\theta_{cap})), \quad (7)$$

$$L_l(\theta_{draw}, \theta_{cap}) = \beta_2(\alpha(\hat{\mathbf{I}}_l, \mathbf{I}^*) - r(\hat{\mathbf{S}}_l)) + (1 - \beta_2)(\beta_1\alpha(\hat{\mathbf{I}}_l, \mathbf{I}^*)) + (1 - \beta_1)L_{mix}(\theta_{cap})), \quad (8)$$

  where $\beta_1, \beta_2 \in [0, 1]$; (7) and (8) are for the CaptionBot being LSTM-XE and LSTM-RL, respectively; $r(\hat{\mathbf{S}}_l)$ is a reward (e.g., the CIDEr metric of the generated sentence $\hat{\mathbf{S}}_l$ w.r.t. the ground-truth sentence $\mathbf{S}^*$); $\alpha(\hat{\mathbf{I}}_l, \mathbf{I}^*)$ is $KL(p(y|\mathbf{I}^*)||p(y|\hat{\mathbf{I}}_l))$ where $p(y|\mathbf{I}^*)$ is the conditional distribution of label $y$ given the ground-truth image $\mathbf{I}^*$ and $p(y|\hat{\mathbf{I}}_l)$ is the conditional distribution of label $y$ given the generated image $\hat{\mathbf{I}}_l$, and $KL(p||q)$ is the Kullback-Leibler divergence from $q$ to $p$; label $y$ is generated by Inception-v3 model in TensorFlow [34]. A label $y$ is selected from 1,000 meaningful classes, such as "Zebra" and "Dishwasher".

- **Step 2: minimizing** $L_r(\theta_{draw}, \theta_{cap})$. As shown in the right hand side of Fig. 1, for a given gold image $\mathbf{I}^*$, the CaptionBot generates a sentence $\hat{\mathbf{S}}_r$; the generated sentence is supplied to the DrawingBot, which produces an image $\hat{\mathbf{I}}_r$. Then we use the loss function $L_r(\theta_{draw}, \theta_{cap})$, which is defined by replacing $(\hat{\mathbf{S}}_l, \hat{\mathbf{I}}_l)$ with $(\hat{\mathbf{S}}_r, \hat{\mathbf{I}}_r)$ in (7) or (8), to calculate the gradients and update $\theta_{draw}$ and $\theta_{cap}$ simultaneously.

- **Step 3:** Go to Step 1 until convergence.

Note that it is critical to choose the right reconstruction loss to enforce cycle consistency in turbo learning. After experimenting with pixel-wise loss, Deep Structured Semantic Models (DSSM) loss [35], and our "perceptual"-like loss $\alpha(\hat{\mathbf{I}}_l, \mathbf{I}^*)$, we found that our "perceptual"-like loss achieves significantly better performance.

It is worth mentioning that the above turbo training is different from GAN training. GAN training is adversarial while our turbo training is collaborative. Actually, our method is more similar to the training of Auto-Encoder (AE). Each branch of the turbo training in Fig. 1) can be viewed as an AE. For example, the left branch is an AE where the DrawingBot is the encoder and the CaptionBot is the decoder. Different from vinilla AE (VAE), our encoding space is semantically meaningful, and supervised signal is available for the encoding procedure because image-caption pairs are available. For the decoder/reconstruction loss, instead of using pixel-wise loss in VAE, we propose to use a "perceptual"-like loss to better capture its semantics.

## 4.4 Experimental results

### 4.4.1 Dataset

To evaluate the performance of our proposed approach, we use the COCO dataset [36]. The COCO dataset contains 123,287 images, each of which is annotated with at least 5 captions. We use the

same pre-defined splits as in [8, 1]: 113,287 images for training, 5,000 images for validation, and 5,000 images for testing. We use the same vocabulary as that employed in [1], which consists of 8,791 words.

Table 1: Performance of CaptionBot and corresponding DrawingBot with BLEU-4 as the reward $r(\hat{\mathbf{S}})$.

| Captionbot | BLEU-4 | CIDEr-D | ROUGE-L | METEOR | SPICE | Drawingbot | Inception |
|---|---|---|---|---|---|---|---|
| Baseline LSTM-XE | 0.2684 | 0.7362 | 0.4937 | 0.2193 | 0.1663 | AttnGAN | 25.68 |
| Turbo LSTM-XE | **0.3168** | **0.7364** | **0.4938** | **0.2196** | **0.1720** | Turbo AttnGAN | **26.69** |
| Baseline LSTM-RL | 0.2831 | 0.7238 | 0.4963 | 0.2190 | 0.1733 | AttnGAN | 25.68 |
| Turbo LSTM-RL | **0.3183** | **0.7327** | **0.4974** | **0.2191** | **0.1735** | Turbo AttnGAN | **26.88** |

### 4.4.2 Evaluation

For the CNN, which is used to extract features from an image, we used ResNet-50 [37] pretrained on the ImageNet dataset. The reason to use ResNet-50 instead of ResNet-152 is mainly for the consideration of training and inference efficiency. We will report the experimental results for ResNet-152 in our future work.

The feature vector $\mathbf{v}$ has 2048 dimensions. Word embedding vectors in $\mathbf{W}_e$ are downloaded from the web [38]. The model is implemented in TensorFlow [34] with the default settings for random initialization and optimization by backpropagation. We empirically set $\beta_1 = \beta_2 = 0.5$.

The widely-used BLEU [39], METEOR [40], CIDEr [32], and SPICE [41] metrics are reported in our quantitative evaluation of the performance of the proposed approach.

Table 1 shows the experimental results on the COCO dataset with 113,287 training samples, for which we use BLEU-4 as the reward $r(\hat{\mathbf{S}})$ in (7), and (8). As shown in the table, our proposed turbo approach achieves significant gain over separate training of CaptionBot and DrawingBot. Following [2], we use the Inception score [42] as the quantitative evaluation measure. The larger Inception score, the better performance. Table 1 shows the Inception score for AttnGAN and Turbo AttnGAN. It is observed that turbo AttnGANs achieve higher Inception scores than AttnGAN.

Table 2 shows the experimental results on the COCO dataset with 113,287 training samples, for which we use CIDEr-D as the reward $r(\hat{\mathbf{S}})$ in (7), and (8). As shown in the table, our proposed turbo approach achieves significant gain over separate training of CaptionBot and DrawingBot.

Table 2: Performance of CaptionBot and corresponding DrawingBot with CIDEr-D as the reward $r(\hat{\mathbf{S}})$.

| Captionbot | BLEU-4 | CIDEr-D | ROUGE-L | METEOR | SPICE | Drawingbot | Inception |
|---|---|---|---|---|---|---|---|
| Baseline LSTM-XE | 0.2684 | 0.7362 | 0.4937 | 0.2193 | 0.1663 | AttnGAN | 25.68 |
| Turbo LSTM-XE | **0.3033** | **0.7473** | **0.4942** | **0.2195** | **0.1721** | Turbo AttnGAN | **26.72** |
| Baseline LSTM-RL | 0.2831 | 0.7238 | 0.4963 | 0.2190 | 0.1733 | AttnGAN | 25.68 |
| Turbo LSTM-RL | **0.3162** | **0.7478** | **0.4981** | **0.2192** | **0.1748** | Turbo AttnGAN | **26.83** |

Because we use ResNet-50 instead of ResNet-152, the performance of the baseline LSTM CaptionBot is not as good as the state-of-the-art LSTM, which uses ResNet-152 or better features.

It is worth mentioning that this paper aims at developing a turbo learning approach to training LSTM-based CaptionBot; therefore, it is directly comparable to an LSTM baseline. Therefore, in the experiments, we focus on the comparison to a strong CNN-LSTM baseline. We acknowledge that more recent papers [11, 12, 13, 14, 1] reported better performance on the task of image captioning. Performance improvements in these more recent models are mainly due to using better image features such as those obtained by region-based convolutional neural networks (R-CNN), or using more complex attention mechanisms [1] to provide a better context vector for caption generation, or using an ensemble of multiple LSTMs, among others. However, the LSTM is still playing a core role in these works and we believe improvement over the core LSTM by turbo learning is still very valuable and orthogonal to most existing works; that is why we compare the turbo LSTM with a native LSTM.

# 5 Semi-supervised learning for CaptionBot

## 5.1 Semi-supervised learning approach for CaptionBot

In this section, we present a semi-supervised learning approach for training CaptionBot.

Human annotation is costly. Hence, many images on the Internet has no caption and semi-supervised learning for CaptionBot is desirable. In fact, the right hand side of Fig. 1 forms a loop, which enables semi-supervised learning for CaptionBot since the closed loop can provide pseudo-labels for unlabeled samples.

As shown in the right hand side of Fig. 1, for a given gold image $\mathbf{I}^*$, the CaptionBot generates a sentence $\hat{\mathbf{S}}$; the generated sentence is supplied to the DrawingBot, which produces an image $\hat{\mathbf{I}}$. The generated image $\hat{\mathbf{I}}$ is supplied to the CaptionBot, which generates a sentence $\tilde{\mathbf{S}}$.

Then we can use the following equation to calculate the loss for an unlabeled sample:

$$L(\theta_{cap}) = \beta_1 \times \alpha(\hat{\mathbf{I}}, \mathbf{I}^*)) + (1 - \beta_1) \times L_{XE}(\theta_{cap}) \qquad \text{for LSTM-XE CaptionBot,} \qquad (9)$$

$$L(\theta_{cap}) = \beta_1 \times \alpha(\hat{\mathbf{I}}, \mathbf{I}^*)) + (1 - \beta_1) \times CIDEr(\hat{\mathbf{S}}, \tilde{\mathbf{S}}) \qquad \text{for LSTM-RL CaptionBot,} \qquad (10)$$

where $\beta_1 \in [0, 1]$; $\alpha(\hat{\mathbf{I}}, \mathbf{I}^*)$ is $KL(p(y|\mathbf{I}^*)||p(y|\hat{\mathbf{I}}))$, and $L_{XE}(\theta_{cap})$ is the cross-entropy between $\hat{\mathbf{S}}$ and $\tilde{\mathbf{S}}$; $CIDEr(\tilde{\mathbf{S}}, \hat{\mathbf{S}})$ is the CIDEr metric of the sentence $\tilde{\mathbf{S}}$ w.r.t. the sentence $\hat{\mathbf{S}}$.

In our experiments, in order to smoothly transit from the initial model trained from labeled data to the model training from labeled and unlabeled data, we adopted the following strategy. For each mini-batch, we use a half number of samples from unlabeled data and another half number of samples from labeled data (sampled from the labeled dataset used to train the initial model). The objective is to minimize the total loss of all labeled and unlabeled samples. We compute the gradient of the total loss of all labeled and unlabeled samples in a mini-batch and update the neural weights $\theta_{cap}$ of the CaptionBot.

Here, we would like to relate our semi-supervised learning to the CycleGAN scheme [43]. Both schemes use cycle consistency, but they are implemented in different ways. Specifically, CycleGAN uses pixel-wise $L_1$ loss while we use "perceptual"-like loss for image semantic consistency. Cycle-GAN has not been applied to text reconstruction, while we use CIDEr score for caption consistency. While CycleGAN was mainly used in homogeneous modality for image-to-image translation, our framework works for multi-modality problems, i.e., image-to-text and text-to-image tasks.

## 5.2 Performance Evaluation

Same as Section 4.4, we also use COCO dataset to evaluate the performance of the proposed semi-supervised learning approach for CaptionBot. We set $\beta_1 = 0.85$.

Table 3 shows the experimental results for semi-supervised learning with 1,000 labeled training samples, 112,287 unlabeled training samples and CIDEr-D as the reward. It is observed that turbo LSTMs significantly outperform baseline LSTMs for semi-supervised image captioning.

Table 3: Performance of CaptionBot with 1,000 labeled training samples, 112,287 unlabeled training samples and CIDEr-D as the reward $r(\hat{\mathbf{S}})$.

| Methods | BLEU-4 | CIDEr-D | ROUGE-L | METEOR | SPICE |
|---|---|---|---|---|---|
| Baseline LSTM-XE | 0.1036 | 0.4331 | 0.3462 | 0.1864 | 0.1231 |
| Turbo LSTM-XE | **0.1832** | **0.6841** | **0.4174** | **0.2071** | **0.1432** |
| Baseline LSTM-RL | 0.1181 | 0.4924 | 0.3518 | 0.1863 | 0.1361 |
| Turbo LSTM-RL | **0.1982** | **0.6954** | **0.4215** | **0.2082** | **0.1466** |

Fig. 4 shows some sample results of CaptionBot and DrawingBot for supervised and semi-supervised learning.

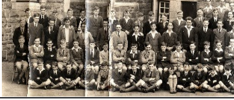



Figure 4: Sample results of CaptionBot and DrawingBot for supervised and semi-supervised learning

## 6 Conclusion

We have presented a novel turbo learning approach to jointly training a CaptionBot and a DrawingBot. To the best of our knowledge, this is the first work that studies both problems in one framework. The framework leverages the duality of image captioning and text-to-image generation and forms a closed loop, which results in a new loss metric by comparing the initial input with the feedback produced by the whole loop. This not only leads to better CaptionBot and DrawingBot by joint training, but also makes semi-supervised learning possible. Experimental results on the COCO dataset have effectively validated the advantages of the proposed joint learning approach.

In our future work, we will explore if adding more unlabeled data can further improve the performance of both bots, and extend the turbo learning approach to other domains, for example, speech recognition vs. text to speech, question answering vs. question generation, search (matching queries to documents) vs. keyword extraction (extracting keywords/queries for documents).

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
