[Supplementary Material]

# CaptionBot: Turbo Supervised Learning

| CaptionBot: Turbo Supervised Learning | CaptionBot: Turbo Supervised Learning | CaptionBot: Turbo Supervised Learning |
|---|---|---|
| **Input image:**  | **Input image:**  | **Input image:**  |
| **Human Generated Sentence**:<br>a grilled cheese sandwich with a side of salad and a pickle | **Human Generated Sentence**:<br>a group of people sitting around a dinner table who just finished eating. | **Human Generated Sentence**:<br>a black and white cat is laying on a green pillow on top of a desk |
| **Baseline LSTM-XE**:<br>a sandwich and a salad on a white plate | **Baseline LSTM-XE**:<br>a man sitting around a table | **Baseline LSTM-XE**:<br>a black and white cat laying on a bed |
| **Baseline LSTM-RL**:<br>a sandwich with a sandwich on a plate | **Baseline LSTM-RL**:<br>a group of people sitting around a table | **Baseline LSTM-RL**:<br>a cat laying on top of a bed |
| **Turbo LSTM-XE**:<br>a white plate topped with a sandwich and a salad | **Turbo LSTM-XE**:<br>a group of people sitting around a table with glasses of wine | **Turbo LSTM-XE**:<br>a black and white cat is sitting on a bed |
| **Turbo LSTM-RL**:<br>a white plate topped with a sandwich and salad on top of a wooden table next to a glass of | **Turbo LSTM-RL**:<br>a group of people sitting around a table eating food at a table in a restaurant | **Turbo LSTM-RL**:<br>a black and white cat laying on top of a bed next to a laptop computer on a desk |

# DrawingBot: Turbo Learning

## DrawingBot generated image

Input Caption: a living room with hard wood floors filled with furniture

**Baseline AttnGAN**          **Turbo AttnGAN**

## DrawingBot generated image

Input Caption: a photo of a homemade swirly pasta with broccoli carrots and onions

**Baseline AttnGAN**          **Turbo AttnGAN**

## DrawingBot generated image

Input Caption: an old clock next to a light post in front of a steeple

**Baseline AttnGAN**          **Turbo AttnGAN**

# CaptionBot: Turbo Semi-Supervised Learning

| CaptionBot: Turbo Semi-Supervised Learning | CaptionBot: Turbo Semi-Supervised Learning | CaptionBot: Turbo Semi-Supervised Learning |
|---|---|---|
| Input image:  | Input image:  | Input image:  |
| **Human Generated Sentence:** a book sitting on top of a wooden desk | **Human Generated Sentence:** a bird perched on top of a wooden power pole | **Human Generated Sentence:** a pizza topped with tomatoes and basil leafs and a glass of cola on a picnic table |
| **Baseline LSTM-XE:** a wooden bench sitting on top of a sidewalk | **Baseline LSTM-XE:** a bird that is standing on a pole | **Baseline LSTM-XE:** a pizza sitting on a plate on a table |
| **Baseline LSTM-RL:** a book on a wooden bench | **Baseline LSTM-RL:** a bird sitting on top of a wire fence | **Baseline LSTM-RL:** a pizza sitting on top of a white plate |
| **Turbo Semi-Supervised LSTM-XE:** a book rest on one of the benches | **Turbo Semi-Supervised LSTM-XE:** a bird perched on top of a tree branch | **Turbo Semi-Supervised LSTM-XE:** a pizza on a white plate on a wooden table and a drink |
| **Turbo Semi-Supervised LSTM-RL:** a travel guide book on a park bench | **Turbo Semi-Supervised LSTM-RL:** a black bird perches at the top of a telephone pole | **Turbo Semi-Supervised LSTM-RL:** a pizza on top of a white plate on a table next to a bottle of wine |