[Reviews · NeurIPS 2018]

Reviewer 1



The authors present a method of jointly training image2text and text2image methods which leads to faster training and better results. The two biggest problems of the paper are from my perspective: - there is no meaningful comparison with the literature. Table 1 and Table 2 have no single comparison result. - the authors should compare their training scheme (which I personally really like) to GAN training. In my view the idea of training two networks "against" each other is very similar in spirit. Further (but beyond the scope of this paper) - I am wondering whether some of the tricks that have recently been invented to make GAN training better/faster/more stable would also apply to the presented method. Similarly, the authors shoudl relate their method for semi-supervised learning to the cyclegan approach Section 3.3: While the method is arguably working well - I don't find the interpretatin in sec 3.3 compelling at all. It would be great to give some more experimental analysis of this point. minor comments: line 19: intensively -> instensely line 18: that requires -> that require line 24: leaving a -> leaving line 39-46: reads as if the two training steps (left/right in Fig1) are performed sequentially - in fact they are performed alternatingly - the authors could make this part much clearer in the paper fig 1: - the visualization of the weight sharing is not really clear - the arrows that go from the bottom back to the top - are not really clear regaring the remaining description of the method the authors refer e.g. to L_r and L_l meaning left part of the figure and right part of the figure. This makes it sometimes hard to follow the notation. If the authors would rewrite this as L_{DC} and L_{CD} (with C for caption, D for drawing) I feel the notation would become easier to follow line 182: is very repetitive - says the same thing twice.

Reviewer 2



Summary: This paper proposed a joint aproach for learning two network: a capitonbot that generates a caption given an image and a drawingbot that generates an image given a caption. For both caption and image generators, the authors use existing network architecture. LSTM - based network that incorporates an image feature produced by Resnet is used for caption generation (the specific architecture is not clearly described). Attention GAN is used to generate an image from caption. The main contribution of this paper is joint training of caption and image generators by constructing two auto-encoders. An image auto-encoder consists of a caption generator feeding an image generator. A caption auto-encoder consists of the same image generator feeding the same caption generator. Both auto-encoders are trained jointly to minimize reconstruction loss of images and captions (in alternative manner). So for example, in the caption auto-encoder, caption generator is trained to generate captions that not only match the ground truth but are also predictive of the underlying concepts in the image. Same holds for training an image generator. This joint learning approach is shown to perform better than independent training of caption and image generators on the COCO dataset. Additionally, authors propose a semi-supervised training modification where a small number of captioned images are used with a large number of uncaptioned ones. The proposed joint (turbo) learning approach results in a caption generator that outperforms an independetly trained one since the latter cannot incorporate unlabeled images. Quality: The idea is interesting and the experimental results validate the advantage of the proposed turbo learning approach. Clarity: The paper is mostly clearly written. However, I did find several places a bit confusing. - When the LSTM-based caption generator is introduced its not clear how the image feature is incorporated in the network. - When the loss for the alternating optimization is explained, I expected it to be a function of the ground truth caption. I guess it’s implicitly buried inside other terms. But it would be good to be more rigorous in your definitions. - Reconstruction loss between a true image and a generated image is measured via KL divergences w.r.t to distribution of object classes found in an image by a pre-trained image classifier. I think this is an important concepts that deserved more emphasis. Originality: While previously explored in Machine Translation, the turbo-learning approach of joint training caption and image generation is novel and interesting. Also similiar “butterfly” architectures have been used in image transfer learning (e.g. Unsupervised Image-to-Image Translation Networks , Lie et al, 2018) Significance: This idea of jointly training two inverse networks can potentially be applied to other domains where such set up is possible.

Reviewer 3



The paper presents a new idea, which is very important in my eyes. The authors propose to jointly train two opposing tasks, in this case an image captioning and image generation model. They prove the effectiveness of their idea, whihc is also of interest for many other applications.